# An Experimental Investigation of Steel Surface Topography Transfer by Cold Rolling

**DOI:** 10.3390/mi11100916

**Published:** 2020-09-30

**Authors:** Dong Xu, Quan Yang, Xiaochen Wang, Hainan He, Youzhao Sun, Wenpei Li

**Affiliations:** National Engineering Technology Research Center of Flat Rolling Equipment, University of Science and Technology Beijing, 30 Xueyuan Road, Haidian District, Beijing 100083, China; yangquan@nercar.ustb.edu.cn (Q.Y.); wangxiaochen@ustb.edu.cn (X.W.); hehainan@ustb.edu.cn (H.H.); sunyouzhao@ustb.edu.cn (Y.S.); 18810918786@163.com (W.L.)

**Keywords:** roughness, transfer, cold rolling, surface topography, reduction rate

## Abstract

Automobile and household appliance panels require steel strips with extremely high-quality surfaces. Therefore, an in-depth study of the surface topography transfer of the steel strip during the rolling process is of considerable significance for improving product quality. In this study, the scale-invariant feature transform (SIFT) algorithm is used to realize the large-field stitching and the correspondence measurement of the surface topography of the roll and strip. The surface topography transfer mechanism and microconvex change law during cold rolling are revealed. Further analysis is conducted regarding the effects of different reduction rates and the initial surface topography of the roll on the formation of strip surface topography. Experimental results reveal that the furrow phenomenon occurs during the rolling process owing to the backward slip effect but is eliminated by the elastoplastic deformation of the matrix and the forward slip action. No furrow occurred along the width direction of the strip. With an increase in the rolling reduction rate, the transfer rate increases, and the strip surface topography is closer to the roll surface topography. Under the same rolling roughness condition and a small reduction rate (5%), the transfer degree increases remarkably with a rise in the reduction rate and increases slowly as the reduction rate continues to grow (from 7 to 10%). This study serves as a theoretical basis for the subsequent improvement of the surface quality of cold rolled strips.

## 1. Introduction

With increasing demand for high-quality surfaces on advanced cold rolled steel strips in the automotive and household appliance industries, the study of the surface quality of the steel strips has become particularly important. The quality of the surface of the cold rolled steel strip directly determines the quality of the product being processed and the downstream processes [1,2,3,4,5,6]. The surface topography is the foundation of the surface quality control of the cold rolled steel strip and is formed on the strip surface during the rolling process [7,8]. Research into the transfer mechanism of the surface topography of cold rolled steel strips has been the focus of scholars. Gorbunov [9] studied the effect of rolling on the surface of cold rolled strips via different treatment methods, comparing the surface amplitude, frequency parameters, and surface correlation of the strip and the spectral and fractal characteristics of the surface micromorphology, showing that the microrelief of the strip is best after being rolled by Topocrom treatment of the roll. Franck [10] used atomic force microscopy (AFM) to perform multiscale roughness measurements to characterize the transfer from steel roll to aluminum alloy strip during the cold rolling of nonlubricating media and 100 μm to 50 nm were investigated over three orders of magnitude of length scales. Roll and strip surfaces have similar self-affine characteristics, and roughness transfer occurs from macro (100 μm) to a very small scale (50 nm). Lenard [11] studied the surface transfer from work roll to strip during cold rolling and analyzed the influence of work roll roughness on other rolling process parameters, concluding that when the roughness of the roll increases to a certain value, the transfer behavior depends on the rolling speed. Ma [12] investigated an experimental study on the effects of surface characteristic transfer and rolling parameters of low carbon steel in cold rolling. The surface characteristics were measured by AFM. The experimental results indicated that the smooth strip surface was rolled under low rolling speed and oil lubrication conditions. Kijima [13,14,15,16] and Shi [17,18] analyzed the influence of roll radius, rolling parameters, and contact conditions on roll roughness transfer through experimental research and numerical simulation. Research revealed that as the diameter of the roll decreases, the peak pressure increases significantly and the transfer of roughness also increases. In addition, Kijima introduced the relationship between rolling force and roughness transfer, and the transfer roughness increased with the rolling force. Ahmed [19] used a three-dimensional contour measurement method for identifying the surface features of cold rolled stainless steel strips for tracking the evolution of pits and roll marks and analyzed how deep pits quickly disappeared and transformed into shallow pits. Bilal [20] and Qu [21] studied the effects of different rolling speeds, reduction rate, lubrication conditions and rolling pressure on rolling transfer efficiency through skin-pass rolling experiments, and obtained the law that the surface roughness and roughness distribution range increase with the increase of reduction rate. Heng [22] investigated changes in microstructure and surface topography of IF steel using electron backscatter diffraction and optical interferometric microscopy. Wu [23] presented a simple approach for analyzing the surface texture transfer in cold rolling of metal strips. Topography characteristics were statistically analyzed based on a large number of field measured data to investigate the roughness failure on textured work rolls and the evolution on steel strips during cold rolling and temper rolling [24]. However, locating the surface micromorphology is difficult because of its small measurement range. In previous works, the experimental selection of the measurement position of the roll surface and the strip surface had a large randomness. Therefore, accurately measuring and contrasting the surface topography of the roll and strip at the corresponding contact position are difficult.

To obtain the three-dimensional microscopic stitching of a large field of view and the measurement of roll and strip correspondence, this paper uses the scale-invariant feature transform (SIFT) feature point matching algorithm. SIFT is an image stitching and matching algorithm commonly used in machine vision and can effectively find feature points in two images and then match the two images. Cong [25] used the SIFT algorithm to experimentally set different thresholds, scaling, rotation, and noise of the image to verify that the algorithm had good robustness and was suitable for rapid and accurate matching. Ran [26] used the SIFT algorithm to collect feature points to achieve regional point matching and realized three-dimensional image reconstruction of the microscopic texture of asphalt pavement. Kai Lin [27] used the SIFT algorithm for face feature matching, and the experimental results demonstrated that the SIFT method can accurately and quickly perform feature matching and effectively reduce the effect of the sharp decline in matching accuracy caused by age changes.

Therefore, in the current work, the surface height information of the sample is converted into a gray image, and the SIFT algorithm is used to realize the stitching and matching of the surface topography of the roll and strip and the corresponding position measurement. This study examines the mechanism of the furrow in the cold rolling process, the transfer mechanism, and the regularity of the surface profile of the roll during cold rolling.

## 2. Experiments and Method

### 2.1. Rolling Experiments

An interstitial-free (IF) steel strip was cut into samples with the measurements of 1.5 mm thick, 30 mm wide, and 70 mm long. The experimental rolling mill is shown in Figure 1a. The diameter of the work roll in the rolling mill was 85 mm and the rolling speed was 15 mm/s in the experiment. The high chromium steel (Cr5) roll of the microexperimental rolling mill was scratched with three marks through electro-discharge texturing [28], in a zone of 2 × 5 cm^2^, roughness 3 μm, 3.5 μm, and 4 μm. Before the cold rolling experiment, the textured surface topography of the roll was copied by the resin. After the cold rolling experiment, the stitching and matching of the surface topography of the roll and strip was carried out by the SIFT algorithm. Then the corresponding position measurements would be realized.

Given that the roll was too large to be measured via a microscope, it was necessary to perform a surface topography operation of the resin copy roll, as shown in Figure 1b before performing the rolling test. JZ-QuickTM ultrafast curing cold buried resin [29] was used to copy the three corners of the roll with a scratched mark to ensure the correspondence of the measurement position. The advantage of the type of resin was its strong micropore gap-filling ability, only a small number of bubbles after curing, and high curing efficiency. Therefore, the surface topography of the roll could be accurately copied. During the use of the resin, a containment wall around the zone to be replicated was created using modeling clay. The resin adhesive and curing agent were mixed at a ratio of approximately 1:1 and poured onto the demarcated zone. The mixture set for 10 min and then was carefully removed from the surface for measurement. The replication was conducted in a fume hood with proper equipment.

After the resin copying step was completed, the surface topography transfer test was conducted under various working conditions with different roll roughness and different reduction rates, using the experimental rolling mill, and the working conditions were set as depicted in Table 1.

### 2.2. Measurement and Characterization of Surface

An Olympus LEXT OLS4100 (Olympus Corporation, Tokyo, Japan) laser scanning confocal microscope (LSCM) was used to observe the surface topography of the resin samples and strips. The microscope magnification was 1200×, which was 0.35 mm away from the sample. The single measurement zone was 258 μm × 258 μm. After the matching process, the corresponding positions of the strip and resin samples were 1200 μm × 1200 μm, and the contact arc zone was 2589 μm × 258 μm.

The calculation of the sample’s statistical parameters to clearly describe its topography features is the following. The surface arithmetic mean height (*S*_a_) is the average value of the deviation of the surface height from the center plane [30]. This surface roughness parameter is universally used because this parameter is easy to determine and easy to measure. The mathematical definition is as follows:(1)$$S_{a}=\frac{1}{MN}\sum_{j=1}^{N}\sum_{i=1}^{M}\left|E\left(x_{i},y_{j}\right)\right|$$
$E\left(x_{i},y_{j}\right)$ is the discrete point of the roughness surface equation.

Roughness transfer ratio (RTR) is the degree to which the surface roughness of the roll is transferred to the surface of the strip. The calculation formula is as follows [13]:(2)$$RTR\left[\%\right]=\frac{S_{a1}-S_{a0}}{S_{\mathrm{ar}}-S_{a0}}\times 100$$
where *S*_a1_ is the surface roughness of the strip after rolling, *S*_a0_ is the surface roughness of the initial strip, and *S*_ar_ is the surface roughness of the roll.

*S*_q_ is the root mean square value of the surface height in the sampling zone to the reference plane, and the mathematical definition is the following.
(3)$$S_{q}=\sqrt{\frac{1}{MN}\sum_{j=1}^{N}\sum_{i=1}^{M}E^{2}\left(x_{i},y_{j}\right)}$$

The surface root mean square deviation *S*_q_ is a parameter that is often used in statistical parameters to indicate the standard deviation of samples, but it does not reflect the distribution and frequency of microscopic surface peaks and troughs.

Skewness (*S*_sk_) is used to measure the symmetry of the surface profile to the reference plane. Surfaces that remove peaks or depth scratches have a negative skewness. Surfaces that are filled with troughs or high peaks have a positive skewness. The mathematical definition is as follows.
(4)$$S_{\mathrm{sk}}=\frac{1}{MNS_{q}^{3}}\sum_{j=1}^{N}\sum_{i=1}^{M}E^{3}\left(x_{i},y_{j}\right)$$

Kurtosis (*S*_ku_) is used to describe the sharpness of the surface probability density, can be used to identify the stability of the surface, and can clearly control the compressive strength of the micro surface. If *S*_ku_ < 3, then the sample surface has few peaks and troughs and the distribution curve is called platykurtic. If *S*_ku_ > 3, then the sample surface has a distribution of high peaks and deep troughs, which is called leptokurtic. The mathematical definition is the following.
(5)$$S_{\mathrm{ku}}=\frac{1}{MNS_{q}^{4}}\sum_{j=1}^{N}\sum_{i=1}^{M}E^{4}\left(x_{i},y_{j}\right)$$

The surface peak density (*S*_ds_) is the number of peaks on the surface zone per unit of the sampling interval, and the calculation formula is as follows:(6)$$S_{\mathrm{ds}}=\frac{N_{s}}{\left(M-1\right)\left(N-1\right)\Delta x\Delta y}$$

$N_{s}$ is the number of peaks of the sampling zone.

Gaussian filtering is performed on the surface topography data of the sample to remove noise before statistical parameter calculation. An assumption is made that the measured original surface topography height is z(x,y), the high frequency roughness signal is r(x,y), and the other low frequency signals are w(x,y). The mathematical model for separating the high frequency surface roughness components is as follows.
(7)r(x,y)=z(x,y)−w(x,y)

A robust discrete Gaussian filtering algorithm is used to determine the datum, its equation:(8)$$w\left(x_{i},y_{i}\right)=\sum_{b=-m_{1}}^{m_{1}}\sum_{d=-m_{2}}^{m_{2}}z\left(\left(i-d\right)\Delta x,\left(j-b\right)\Delta y\right)\cdot s\left(d\Delta x,b\Delta y\right)\cdot \Delta x\Delta y$$
where $m_{1}$ and $m_{2}$ are the half window widths of the Gaussian filter weight function in the x and y directions, respectively.

### 2.3. Surface Microtopography Stitching and Matching Based on Scale-invariant Feature Transform (SIFT) Algorithm

The data processing based on the SIFT algorithm converts the height information of the sample surface data into picture information: it performs gray processing on each pixel point, it identifies the key points of the two pictures through translation, rotation, and zoom, and determines the feature vector. The nearest neighbor and the next nearest neighbor algorithms are used to calculate the Euclidean distance between the key points, which is then compared with the set threshold. The feature points are lastly determined by judging the same point of the gray gradient change, and then the stitching is performed. Figure 2a,b show the experimentally obtained resin and strip height data, which were converted into grayscale image format, as presented in Figure 2c,d. The feature points were determined via the SIFT algorithm.

A stitching process must be performed to obtain a detailed image of a wide range of complete surface topography information. Several measurements were taken prior to stitching, and each measurement had a certain overlap with the previous one. The feature points of the overlap region were found through the SIFT algorithm, and the feature points were stitched [31]. After stitching all the images, a large image was created as the stitched complete image, as illustrated in Figure 2e. The data obtained by LSCM usually have certain errors, such as noise, outliers, and dead pixels. Gaussian filtering was performed prior to stitching and matching to minimize these erroneous data.

The large-field image information of the resin and the strip samples obtained after the matching were again processed via the SIFT algorithm to find the corresponding feature points, as depicted in Figure 2f. Furthermore, the corresponding positions of the two samples’ surface topography were obtained.

## 3. Results and Discussion

### 3.1. Roller and Strip Initial Surface Topography

Figure 3a–c show the surface topography of the resin samples measured by LSCM and the Figures also depict the microscopic topography surface of the roll after electric discharge texturing. Figure 3d shows the surface topography of the untested strip.

The surface parameters of the rolls and strips were calculated and shown in Table 2. The initial surface roughness of the strip was much smaller than the roll roughness. Therefore, the influence of the surface topography of the strip on the rolling transfer was negligible.

Gaussian filtering was performed on the four surface measurement zones, and the filtered data were subjected to normal probability density distribution processing at height, as revealed in Figure 4. The smaller the roughness, the more concentrated the surface height distribution. As the roughness increased, the difference in height distribution increased.

### 3.2. Surface Topography of Strip Arc Contact Zone

Considering that the morphological changes of the contact arc zone were basically the same under the nine working conditions, this section selected the roll with the roughness of 3.5 μm and the strip rolled with the reduction rate of 7% as the research object. Figure 5 presents the surface topography of the contact arc zone. Figure 5a,c are 2D images of the contact arc zone. Figure 5b,d are 3D topographic images.

The length of the complete contact arc was 2590 μm, which was divided into five zones every 518 μm. Figure 5 reveals that Zones IV and V were not in contact with the roll, and the surface roughness was small. Zone III was in contact with the roll and the roughness of the surface began to increase. The roughness of Zones I and II was stable with the deepening of the rolling. The surface parameters of the five zones were calculated separately to obtain the surface features of the contact arc region, and the variation along the length direction was studied. Table 3 presents the surface parameters of each part of the contact arc zone.

The *S*_a_ in Table 3 is a good description of the change in the surface roughness of the strip during the rolling process. Figure 6 reveals that the *S*_a_ of the strip enlarged as the rolling progressed and the value of *S*_a_ gradually stabilized. According to the surface skewness (*S*_sk_) and the kurtosis (*S*_ku_) in Table 3, the surface parameter distribution of the strip during rolling was random, and the skewness and kurtosis of the strip were consistent with the distribution of the roll surface when the rolling was stable. The numerical value of the surface peak density in Table 3 explains that the surface peak density value was too large during the rolling process. As the rolling was stabilized, the surface peak density was close to the surface peak density of the roll completely copied.

A 2D surface of the contact arc zone along the rolling direction and the width direction was selected for the study to clearly understand the morphology change law of the contact arc zone during the rolling process, as illustrated in Figure 7a,c. The black curve was the 2D shape profile on the contact arc zone, and the red curve was the Gaussian filter midline. The contact arc along the rolling direction was divided into three regions by the rolling principle, namely: I backward sliding region, II intermediate region, and III frontward sliding region, as shown in Figure 7a. Through observation and analysis, the surface topography of the three regions was simplified to that in Figure 7b. The strip was in contact with the roll, and the rolling entered the backward sliding area. At this time, the strip and the roll had not yet produced relative displacement. However, the speed difference between the two was the largest at this point, and the strip speed was backward relative to the roll. Over time, the convexity on the roll was copied onto the strip, and the strip was moved backward, thereby resulting in a small slope on the right side of the corresponding groove and a large slope on the left side. After rolling into the intermediate zone, the horizontal linear velocity of the roll and the strip were basically the same, and no relative sliding could be considered. The fluidity of the metal was added to the side with the smaller slope on the right side of the pit, therefore the right-side slope was gradually increasing. When entering the frontward sliding area, the strip moved forward relative to the roll, and the slope of the left side of the pit continued to be very large. On the right side, owing to the filling of the metal fluidity and the influence of the roll peaks, the slope was gradually increased. Lastly, the slopes on both sides of the pit were equivalent and the entire pit was horizontally oriented. Figure 7c is a 2D surface contour randomly selected from the width direction in the intermediate region. The Figure shows that during the rolling process, the contact arc zone did not have a furrow phenomenon along the width direction, which explains the change process of the surface topography during the strip rolling. After the furrow was generated, the furrow was gradually filled and finally disappeared with the rolling progress. This slope change was caused by the influence of the frontward and backward slip and the elastoplastic deformation of the steel strip matrix. In addition, Figure 7a reveals that the normal directions of the pits were at an angle with the direction of the rolling force, perpendicular to the horizontal direction.

### 3.3. Surface Topography of the Strip after Cold Rolling

The 2D surface topography of the strip after rolling was obtained, as depicted in Figure 8a,c,e. Figure 8b,d,f show the 3D topography of the strip after rolling. The 3D topography revealed that under the same roll roughness, as the reduction rate increased, the surface of the strip had a deeper pit and a deeper peak.

SIFT technology was used to match the processed resin and strip data to determine the corresponding positions. Figure 9a illustrates that the roll roughness was 3.5 μm and the reduction rate was 7%. The 2D surface topography of the resin and strip along the rolling direction was randomly selected. The resin was the result of complete copying of the roll, therefore, the 2D profile of the resin could be regarded as the 2D topography of the roll. Figure 9a shows that the deep trough of the roll could not be transferred 100% to the strip surface, and the transfer rate was low. For the gentler area of the trough, the surface profile of the roll was transferred to the strip. The efficiency was high, the peak of the roll corresponded to the trough of the strip, and the roll in this area was generally lower than the trough of the strip, because the peak of the roll was in contact with the surface of the strip during the rolling process. The peak indentation caused plastic deformation of the strip, and the metal on the surface of the strip mainly flowed downward to form a pit of a similar shape. Transfer to the strip is shown in Figure 9b. As the rolling of the roll peaks was deepened to the extent of the strip, the pits of the strip were deepened, and the surrounding metal was generated to bulge upward while part of the metal flowed upward to fill the roll pits. Fully copying the pit shape on the roll was difficult because of the metal flow restrictions, as shown in Figure 9c. When the roll left the surface of the strip, the rebound of the strip surface caused by the elastoplastic deformation principle of the metal caused the peak of the strip to be slightly higher than the trough of the roll, and the trough of the strip was slightly higher than the peak of the roll, as illustrated in Figure 9d.

The 2D topography of the strip rolled under different reduction rate and the roll were randomly selected in the rolling direction, as shown in Figure 10a,c,e. The normal probability density distribution at height were performed, as revealed in Figure 10b,d,f. As the reduction rate increased, the roughness distribution range of the strip increased and approached the roll roughness distribution range. The 2D topography along the width direction is depicted in Figure 11a,c,e. The normal distribution was performed, as revealed in Figure 11b,d,f. The roughness transfer rate in the rolling direction increased as the reduction rate increased. However, the roughness transfer rate in the width direction did not change substantially with the change in the reduction rate, and the roughness transfer was kept at a very high value. This finding was caused by the poor ductility of the strip in the width direction during the rolling process and by the ease of copying the surface topography of the roll. The strip steel flowed forward in the rolling direction, and the ability to fill the roll was poor. The roughness transfer was low under the low-reduction rate condition, and the roughness transfer enlarged as the reduction rate increased.

The surface parameters of the samples under various working conditions were calculated. The surface parameters *RTR*, *S*_sk_, *S*_ku_, and *S*_ds_ are shown in Figure 12. In Figure 12a, where the roll roughness was the same, the reduction rate was increased from 5 to 7%, and the transfer ratio increase range was from 30 to 40%. In short, when the reduction rate was small, the surface roughness of the strip increased rapidly as the reduction rate increased, and the surface topography of the roll could be quickly copied to the strip. However, as the reduction rate increased from 7 to 10%, the transfer ratio was increased by about 10%, the surface roughness of the strip increased slowly. Figure 12b reveals that the profiles of the strip and the corresponding roll had negative skewness, that is, profiles with peaks removed or deep scratches. At the same reduction rate, the skewness of the strip was affected by the skewness of the roll. In Figure 12c the kurtosis of the strip was larger than 3 under the condition of the reduction rate of 5% regardless of the roll roughness. At the condition of the reduction rate of 5%, it could be seen that the kurtosis decreased with the increase of the *S*_a_ of the roll. As the *S*_a_ of the roll increased, the kurtosis of the strip was smaller than 3 and close to the value of the roll under the conditions of 7 and 10% reduction rate, meaning the strip and rolls had similar peaks and troughs, all of which were platykurtic. With the increase in the rolling reduction rate, the sharpness of the probability density of the strip surface had the same distribution as the roll. From Figure 12d, as the rolling reduction rate increased, the surface peak density became increasingly small, and the peak density of the strip surface was inclined to the roll. The rolling reduction rate could be inferred to have had an effect on the surface peak density during the rolling transfer process. At the same reduction rate, it could be seen that the strip surface peak density increased with the increase of the *S*_a_ of the roll. The roll parameters could be found having an influence on the surface topography transfer.

Based on the trend and regularity of the aforementioned surface parameters, the mechanism phenomenon existing in the rolling transfer process was analyzed. During the rolling process, the microprotrusions on the surface of the roll could be rapidly pressed into the strip to deform the strip substrate, but the ability of the strip to fill the surface profile of the roll was affected by the reduction rate. When the reduction rate was small, the surface profile of the strip filling roll was easy, and as the reduction rate increased, the filling of the roll profile increased rapidly. As the reduction rate was further increased, the surface metal of the strip needed to fill more of the surface of the roll in addition to requiring increased force to move downward to make the strip thinner. At this time, as the reduction rate increased, the filling of the surface profile of the roll slowly increased until the surface profile was nearly completely filled, and the strip and the roll had similar surface topographies.

## 4. Conclusions

(1) The experimental results reveal that the direction of the pits in the rolling contact arc zone is different from the direction of the pits described in the rolling principle, which should be along the direction of the rolling force. The pits in the contact arc zone are basically perpendicular to the rolling direction and form a certain angle with the direction of the rolling force. During the rolling process, the furrow phenomenon occurs in the rolling direction due to the backward slip. With the elastoplastic deformation of the strip substrate and the forward slip action, the furrow gradually becomes smaller until it disappears. No furrow occurs along the width direction of the strip.

(2) Experiments show that the transfer efficiency of the roll surface topography transferred to the strip during cold rolling is affected through the roll roughness and rolling reduction rate. When the roll roughness is constant and the reduction rate is increased from 5% to 7%, the transfer ratio is increased by approximately 40% and increases rapidly. However, when the reduction rate is increased from 7% to 10%, the transfer ratio is only increased by approximately 10% and increases slowly. With an increase in the rolling reduction rate, the distributions of the skewness and kurtosis of the strip surface become increasingly close to the roll, and the surface peak density approaches the roll peak density. In short, with an increased reduction rate, the surface topography of the strip becomes approximately the same as the surface topography of the roll.

(3) After the roll leaves the strip surface, the elastic recovery of the strip causes the groove portion of the strip to be slightly higher than the convex portion of the corresponding roll because of the elastoplastic deformation of the metal.

## Figures and Tables

**Figure 1 micromachines-11-00916-f001:**
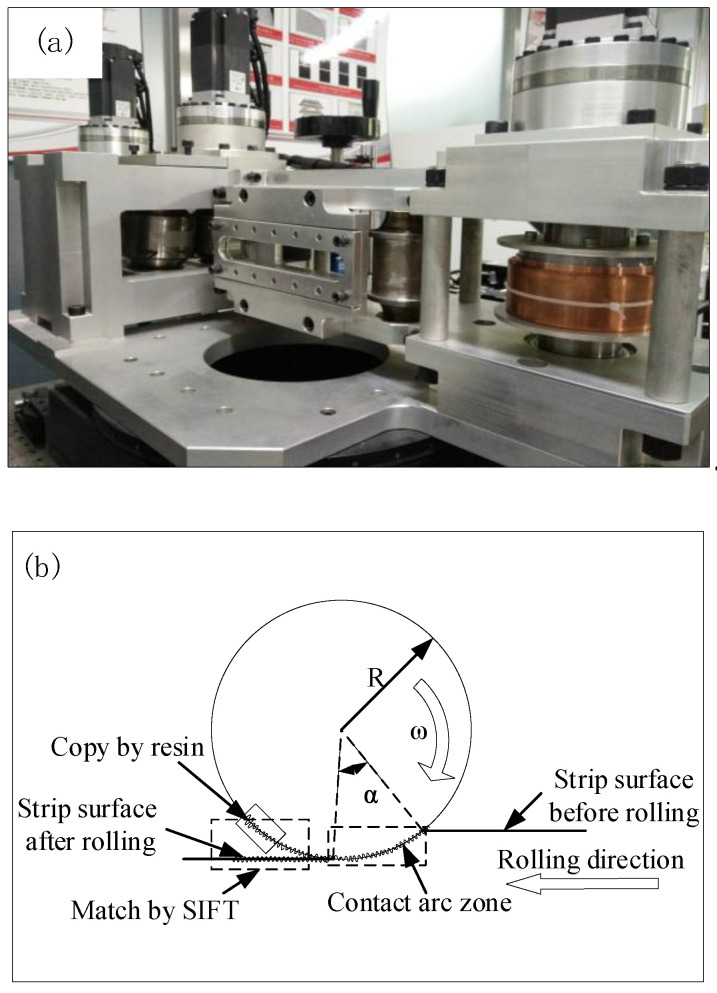
Experimental equipment. (**a**) Rolling mill. (**b**) Rolling process.

**Figure 2 micromachines-11-00916-f002:**
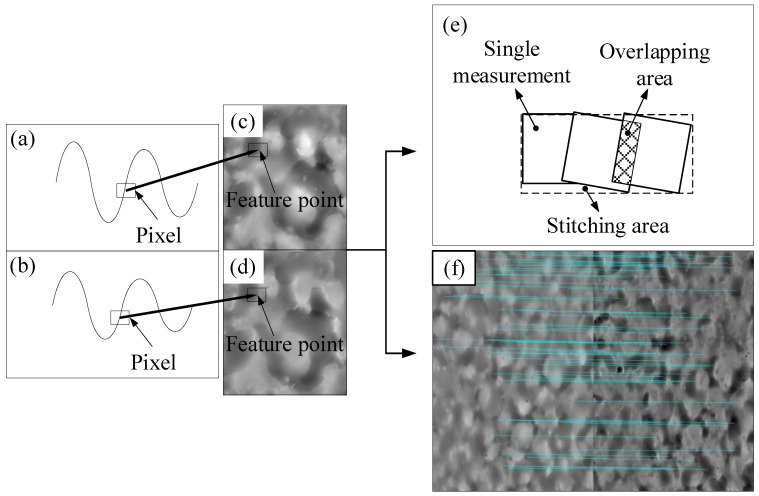
Scale-invariant feature transform (SIFT) matching and stitching process. (**a**) Resin height data, (**b**) strip height data, (**c**) grayscale image of resin height data, (**d**) grayscale image of strip height data, (**e**) the stitched complete image, (**f**) the corresponding feature points between the resin and strip surfaces.

**Figure 3 micromachines-11-00916-f003:**
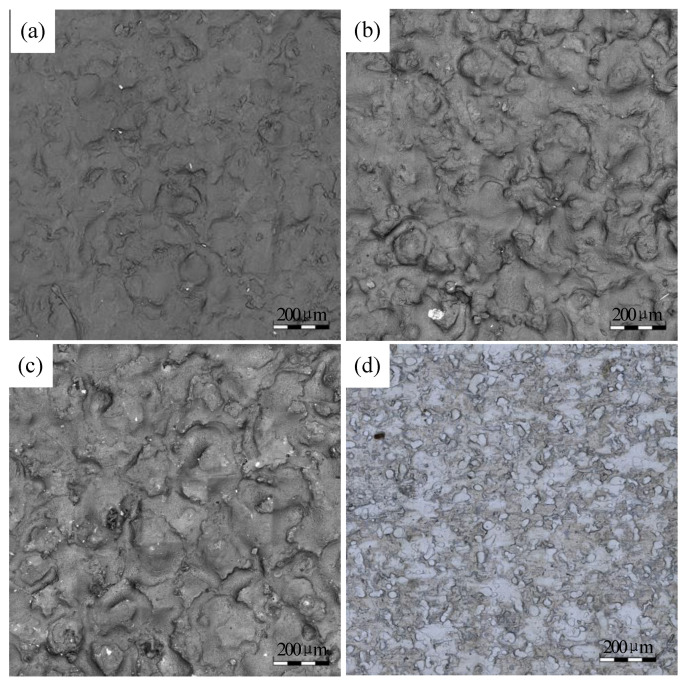
Initial surface topography: (**a**) the *S*_a_ of roll 3 μm, (**b**) the Sa of roll 3.5 μm, (**c**) the *S*_a_ of roll 4 μm, (**d**) the strip before rolling.

**Figure 4 micromachines-11-00916-f004:**
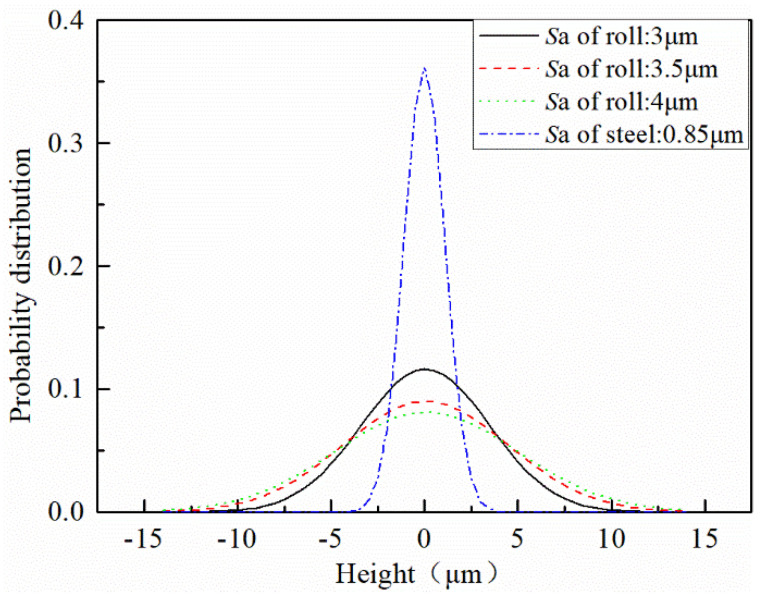
Probability distribution of roll and strip surface height.

**Figure 5 micromachines-11-00916-f005:**
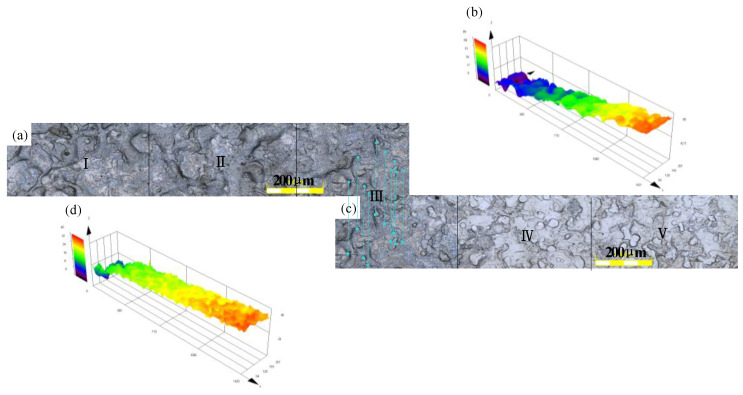
Surface topography of contact arc zone. (**a**,**c**) The 2D LSCM scan images. (**b**,**d**) The 3D LSCM scan images.

**Figure 6 micromachines-11-00916-f006:**
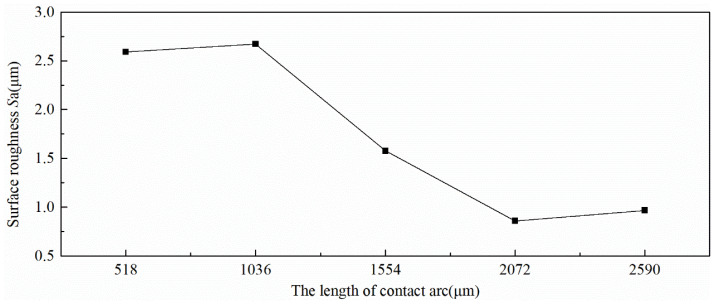
Change of the *S_a_* in contact arc zone.

**Figure 7 micromachines-11-00916-f007:**
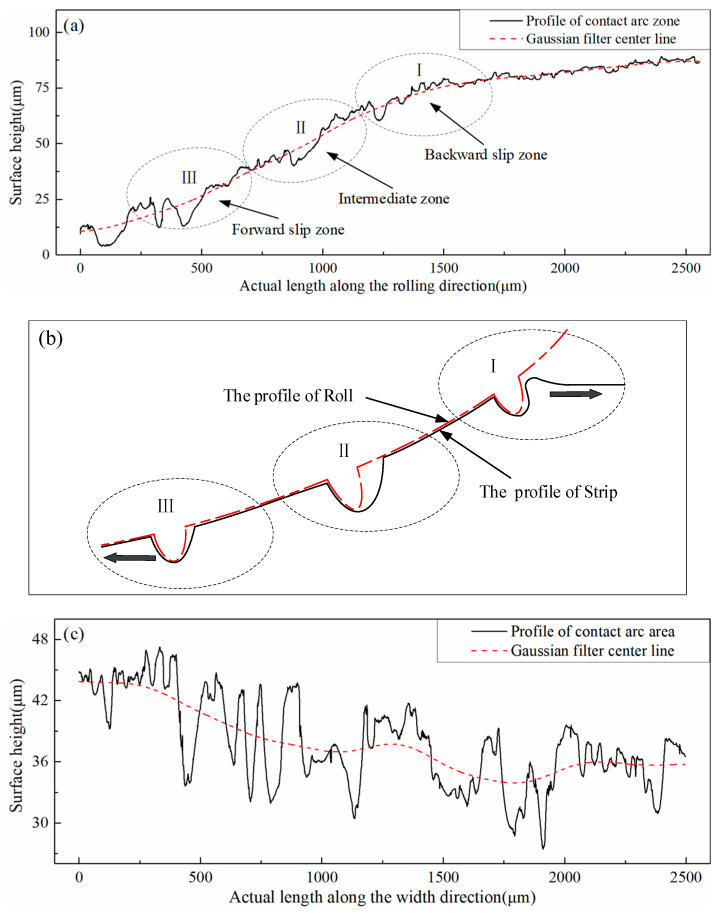
Surface topography of strip contact arc zone. (**a**) A 2D profile of the strip along the rolling direction. (**b**) Furrow variation in contact arc zone. (**c**) A 2D profile of the strip in the intermediate zone (Ⅱ) along the width direction.

**Figure 8 micromachines-11-00916-f008:**
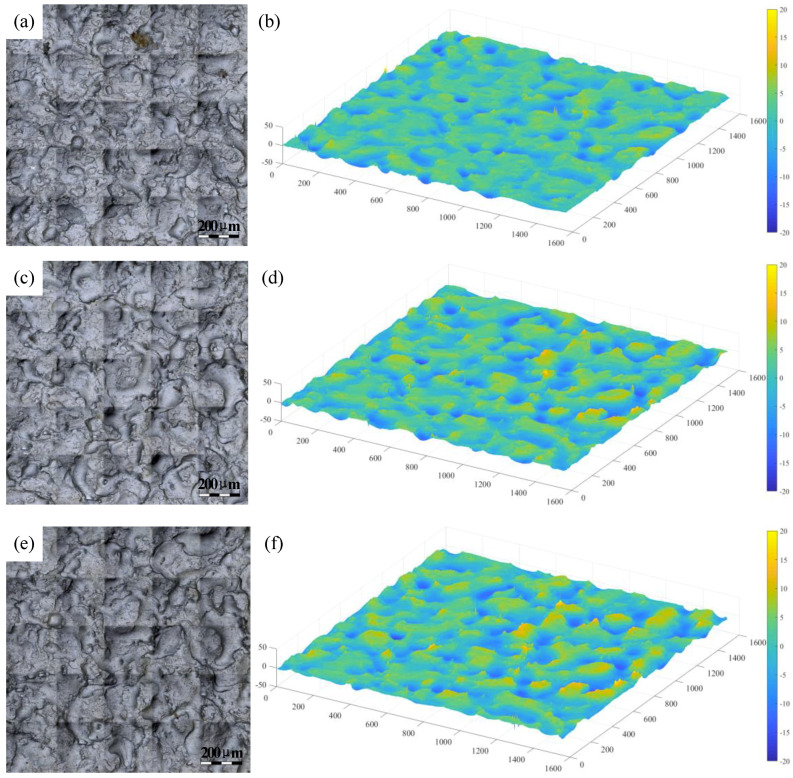
Strip surface topography after rolling (the Sa of roll 3.5 μm), (**a**) and (**b**) 5% reduction, (**c**) and (**d**) 7% reduction, (**e**) and (**f**) 10% reduction.

**Figure 9 micromachines-11-00916-f009:**
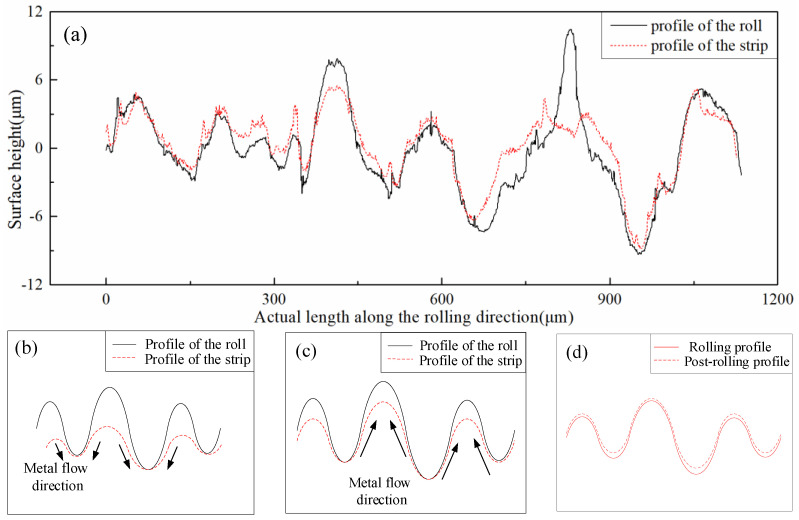
Mechanism of strip surface topography during the rolling process: (**a**) profile of roll and strip along the rolling direction, (**b**) initial press-in process, (**c**) strip filling process and (**d**) strip elastic recovery.

**Figure 10 micromachines-11-00916-f010:**
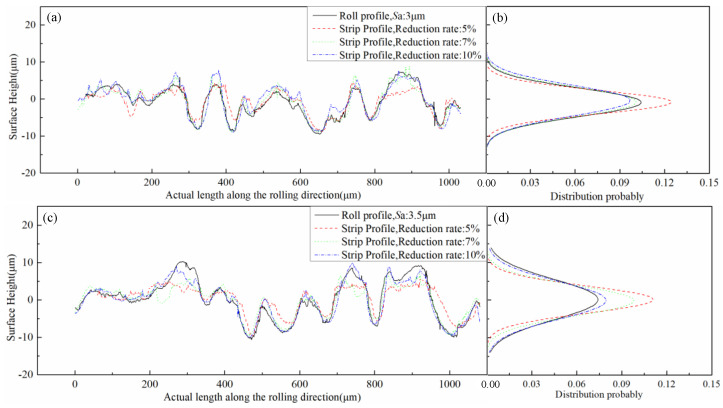
Surface topography of roll and strip under different roll roughness and reduction rates along the rolling direction: (**a**) profile of roll and strips (3 μm roll roughness, 5/7/10% reduction), (**b**) height distribution of roll and strips (3 μm roll roughness, 5/7/10% reduction), (**c**) profile of roll and strips (3.5 μm roll roughness, 5/7/10% reduction), (**d**) height distribution of roll and strips (3.5 μm roll roughness, 5/7/10% reduction), (**e**) profile of roll and strips (4 μm roll roughness, 5/7/10% reduction), (**f**) height distribution of roll and strips (4 μm roll roughness, 5/7/10% reduction).

**Figure 11 micromachines-11-00916-f011:**
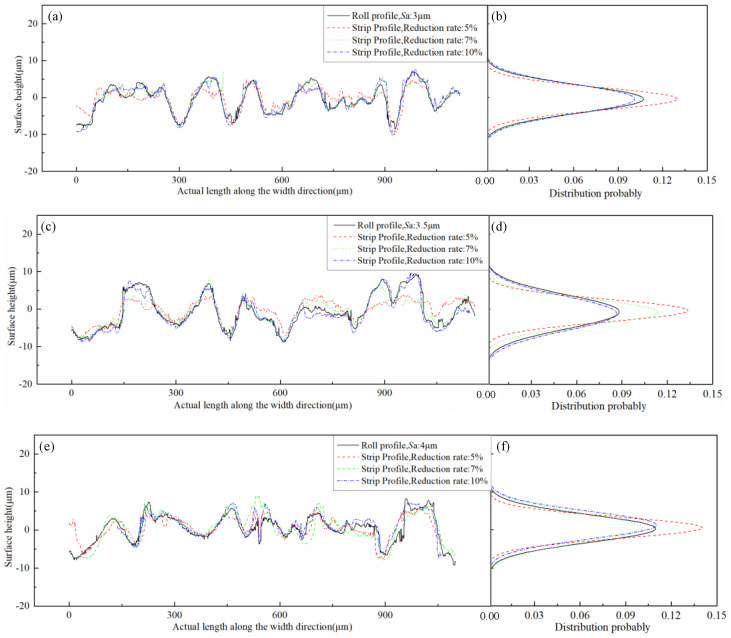
Surface topography of roll and strip under different roll roughness and reduction rates along the width direction: (**a**) profile of roll and strips (3 μm roll roughness, 5/7/10% reduction), (**b**) height distribution of roll and strips (3 μm roll roughness, 5/7/10% reduction), (**c**) profile of roll and strips (3.5 μm roll roughness, 5/7/10% reduction), (**d**) height distribution of roll and strips (3.5 μm roll roughness, 5/7/10% reduction), (**e**) profile of roll and strips (4 μm roll roughness, 5/7/10% reduction), (**f**) height distribution of roll and strips (4 μm roll roughness, 5/7/10% reduction).

**Figure 12 micromachines-11-00916-f012:**
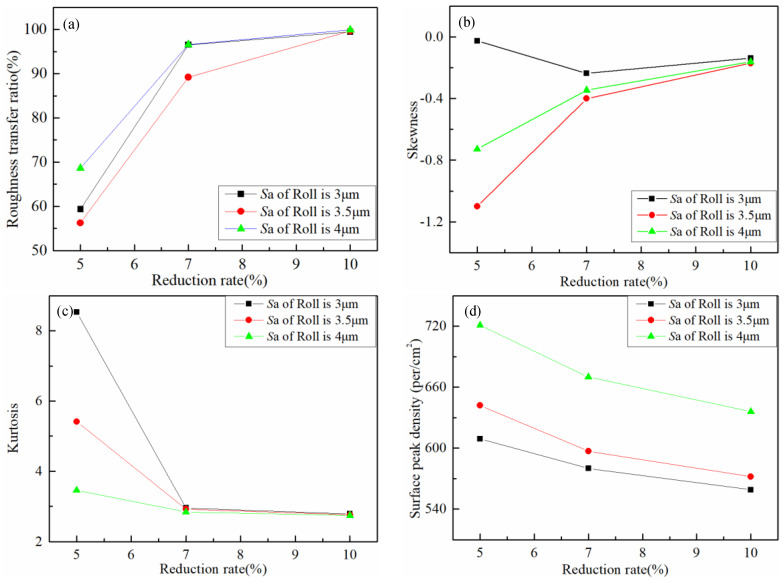
Strip surface parameters under different conditions: (**a**) roughness transfer ratio, (**b**) skewness, (**c**) kurtosis and (**d**) surface peak density.

**Table 1 micromachines-11-00916-t001:** Working condition settings.

*S*_a_ of Work Roll (μm)	Reduction Rate
3 μm	5%
3.5 μm	7%
4 μm	10%

**Table 2 micromachines-11-00916-t002:** The roughness parameters of roll and initial strip.

Specimen	*S*_a_ (μm)	*S* _sk_	*S* _ku_	*S*_ds_ (per/cm^2^)
A(Resin)	2.99	−0.004	2.821	556
B(Resin)	3.50	−0.045	2.853	569
C(Resin)	4.07	−0.036	2.958	636
D(strip steel)	0.85	−0.036	3.591	606

**Table 3 micromachines-11-00916-t003:** Surface parameters of the contact arc along the length.

The Number of Part	*S*_a_ (μm)	*S* _sk_	*S* _ku_	*S*_ds_ (per/cm^2^)
1	2.59	−0.376	2.699	574
2	2.67	−0.813	3.099	596
3	1.57	−0.731	4.506	677
4	0.86	−0.072	2.602	634
5	0.97	−0.200	2.555	615

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
