# Peer review of "An Experimental Investigation of Steel Surface Topography Transfer by Cold Rolling"

_micromachines, 2020, doi:10.3390/mi11100916_

Round 1
Reviewer 1 Report
A very interesting article, the amount of research and analysis of results allows for a comprehensive approach to the topic. The transparency of presentation and the selection of research and analysis methods significantly facilitate the reception of research results. A synthetic approach made it possible to present reasoned conclusions. My questions concern only the rollers used, please attach a short description of the researcher's stand, including the description of the surface quality in the work rollers and the methodology of their improvement. In Figure 1, photos are missing (and it should be at the top), moreover, despite the fact that Figure 1b is only a diagram, my questions concern the kink and adhesion of the tape to the roller, whether the tape is pressed by additional tools, if not how the process in Figure 1b is carried out, please however, it is quite confusing for the audience. Technical Note from line 340 we have a summation with a summary which starts with line 350 in the text, which should be corrected.Author Response
Response to Reviewer 1 Comments
Point 1: My questions concern only the rollers used, please attach a short description of the researcher's stand, including the description of the surface quality in the work rollers and the methodology of their improvement.
Response 1: This study examines the mechanism of the furrow in the cold rolling process, the transfer mechanism, and the regularity of the surface profile of the roll during cold rolling. In the actual production process, according to the demand for the surface topography of the product, the original surface topography of the work roll and reduction rate can be chosen by the obtained regularity.
Point 2: In Figure 1, photos are missing (and it should be at the top), moreover, despite the fact that Figure 1b is only a diagram, my questions concern the kink and adhesion of the tape to the roller, whether the tape is pressed by additional tools, if not how the process in Figure 1b is carried out, please however, it is quite confusing for the audience.
Response 2: Considering the reviewer’s suggestion, Figure 1: (a) is placed in front of (b), and annotations of the test rig are added to the text (Line89-95 highlighted in green). The high chromium steel (Cr5) roll of the micro-experimental rolling mill (Fig. 1 (a)) was scratched with three marks through electro-discharge texturing, which zone is 2×5 cm2, roughness is 3 μm, 3.5 μm, and 4 μm.
Point 3: Technical Note from line 340 we have a summation with a summary which starts with line 350 in the text, which should be corrected.
Response 3: The Technical Note from line 340 to line 349, is discussion and explanation of Fig.12. Therefore, we think, it is more appropriate to put it in Section 3(Results and discussion).

Reviewer 2 Report
Comments on micromachines-942693-peer-review-v1
This paper presents an experimental study on surface topography transfer mechanism and micro-convex change law in cold rolling. The reviewer believe that some comments should be addressed before publication.
1 The English writing could be improved and polished to a more standard level.
2 Recently, many good articles about wear and material transfer have been published in 2019/2020, and some of them could be referred and cited.
3 Figure 1: (a) should be placed in front of (b). Annotations are necessary when displaying the test rig.
4 Could the authors give more info about the blue (green) lines in figure 2 (f)?
5 Any transfer theory could be compared or validated by the experimental results? Or any theory could be developed based on the results?
Author Response
Response to Reviewer 2 Comments
Point 1: The English writing could be improved and polished to a more standard level.
Response 1: We have modified the English expression as the suggestion from Reviewer, which are highlighted (yellow) in the revised version.
Point 2: Recently, many good articles about wear and material transfer have been published in 2019/2020, and some of them could be referred and cited.
Response 2: The comment is very valuable and helpful for improving our paper. Some related References on material transfer published in 2018-2020 have been added (Line54-63 highlighted in green)
Point 3: Figure 1: (a) should be placed in front of (b). Annotations are necessary when displaying the test rig.
Response 3: Considering the Reviewer’s suggestion, Figure 1: (a) is placed in front of (b), and annotations of the test rig are added to the text (Line89-95 highlighted in green).
Point 4: Could the authors give more info about the blue (green) lines in figure 2 (f)?
Response 4: The blue (green) lines connect the corresponding SIFT feature points in in the resin and the strip surface topography. The corresponding SIFT feature points could be used to realize the matching of the surface topography of the roll and strip.
Point 5: Any transfer theory could be compared or validated by the experimental results? Or any theory could be developed based on the results?
Response 5: The previous studies focus far more on the statistics of the surface topography (surface roughness, for instance). Furthermore, Lack of effective means for corresponding position measurement at the micro scale makes it impossible to quantify the press-in and furrow phenomenon at the cold rolling interface. In the current work, the SIFT algorithm is used to realize the stitching and matching of the surface topography of the roll and strip and the corresponding position measurement. This study examines the mechanism of the furrow in the cold rolling process, the transfer mechanism, and the regularity of the surface profile of the roll during cold rolling.

Reviewer 3 Report
Dear authors,
I am sure that your manuscript with the title “An experimental investigation of steel surface transfer by cold rolling” will be interesting for both academic and industrial communities. The problem related ship with the surface quality of cold-rolled steels is very actual nowadays, namely in the automotive industry.
Questions and remarks:
- All pictures in Figure 2 must be described in the title of Figure.
- The scales in Figure 3 are unreadable. Please improve the quality of scale (benchmark) on these pictures of topography.
- The rolling process must be more detailed described. Which diameter of the roll used in the experiment, which rolling rate of roll used? These parameters also influence on the roughness of cold rolling steels.
- Normally, different types of lubricants used for the cold rolling processes that decrease friction and improve the roughness of the surface of steels. Why lubricant did not use in this study?
- If this work is focused on the investigation of the surface roughness of cold rolling steels after the different reduction of thickness, why the initial strips characterized by some roughness? It would be better if authors carried out and include in the manuscript the experiment with strips that were precise polishing before the rolling and then compare the obtained results.
Best regards.
Author Response
Response to Reviewer 3 Comments
Point 1: All pictures in Figure 2 must be described in the title of Figure.
Response 1: The description of pictures in Figure 2 has been added (Line183-185 highlighted in green).
Point 2: The scales in Figure 3 are unreadable. Please improve the quality of scale (benchmark) on these pictures of topography.
Response 2: The quality of scales in Figure 3,5,8 have been improved so that they can be seen clearly.
Point 3: The rolling process must be more detailed described. Which diameter of the roll used in the experiment, which rolling rate of roll used? These parameters also influence on the roughness of cold rolling steels.
Response 3: The comment is very valuable and helpful for improving our paper. The diameter of the work roll in the rolling mill is 85mm and the rolling speed is 15mm/s in the experiment. Some more detailed descriptions have been added to the text (Line89-95 highlighted in green)
Point 4: Normally, different types of lubricants used for the cold rolling processes that decrease friction and improve the roughness of the surface of steels. Why lubricant did not use in this study?
Response 4: In industrial production, there are two modes of cold rolling in temper rolling process, with lubricant and without lubricant. This work is focused on the surface topography transfer without lubricant. Lubrication at the interface makes the mechanism of the surface topography transfer more complicated because of the mixed lubrication. Further research will be carried out in the following research.
Point 5: If this work is focused on the investigation of the surface roughness of cold rolling steels after the different reduction of thickness, why the initial strips characterized by some roughness? It would be better if authors carried out and include in the manuscript the experiment with strips that were precise polishing before the rolling and then compare the obtained results.
Response 5: As the reviewer said, the experiment with strips that were precise polishing before the rolling will obtain more accurate results. The initial surface morphology of the steel samples is original state after cold rolling without artificial special processing. The initial surface roughness of the strip is much smaller than the roll roughness. Therefore, the influence of the surface morphology of the strip on the rolling transfer is negligible.
